# GeneDAE: A Sparse Denoising Autoencoder for Deriving Interpretable Gene Embeddings

**Monica Isgut,**[*] **Neha Jain,**[*] **Andrew Hornback, Karan Samel & May D. Wang**
Georgia Institute of Technology

## Abstract

A challenge in genomics research involves identifying functionally relevant genes associated with diseases. We present GeneDAE, a sparse denoising autoencoder that extracts gene representations from large-scale population-level genotype data, which can then be used to identify gene-to-disease associations. The GeneDAE encoder and decoder connections are modeled on a bipartite biological knowledge graph that connects individual variants (single nucleotide polymorphisms; SNPs) to their nearby genes, enabling each node in the hidden layer to be used as an interpretable, multi-purpose gene embedding derived using information only from variants in close proximity that are most likely to impact gene function. We use the UK Biobank dataset and focus on the major histone compatibility complex (MHC) region of the genome, which is critical to immune function and autoimmune disease pathophysiology. Using GeneDAE, we extracted 239 MHC gene embeddings and identified novel gene-to-disease associations.

## 1 Introduction

An ongoing area of investigation in genomics involves identifying the most important genes functionally associated with a given disease. Genome-wide association studies (GWAS) have been instrumental for identifying important genomic variants (i.e., genotypes, single nucleotide polymorphisms; SNPs) associated with a disease of interest. While some variants are located within gene coding regions and are thus clearly associated with the gene, more than 90% of variants are located outside coding regions (Cano-Gamez & Trynka, 2020), making it challenging to identify the genes associated with those variants that contribute towards a given trait or disease. To address this challenge, we present GeneDAE, a sparse denoising autoencoder designed to extract gene representations from large-scale population-level genotype data. Our approach uses an adjacency matrix to constrain the connectivity of the encoding and decoding layers based on variant-to-gene annotations connecting variants to nearby genes. This enables extraction of interpretable gene embeddings, whereby information going into each hidden node only contains data from variants located in close proximity to the gene, more likely to be functionally relevant. These gene embeddings can then be used in downstream tasks for disease or trait prediction and to discover gene-to-disease associations.

Our work builds on previous work by Ma et al. (2018) (DCell), which first introduced the notion of using biological knowledge graphs to constrain the connectivity of a neural network by pruning its weights based on known biological connections between concepts. The work on DCell showed that forcing sparsity in this way enables hidden nodes in a neural network to be interpreted in terms of the biological concepts they represent in the knowledge graph, while maintaining similar performance to that of a fully connected network. DCell was used to predict yeast cell growth, and GenNet (van Hilten et al., 2021) explored this framework in the context of human population genomics. Unlike previous work, GeneDAE utilizes these concepts in an autoencoder to extract multi-purpose embeddings of genes that can then be used in downstream applications, alone or in combination with other data modalities. Since genotype data can be extremely high-dimensional, sometimes with millions of features, a limitation of existing work involves the extensive computational resources needed for model training. GeneDAE enables gene embeddings to be derived independently for each gene or gene set of interest and decouples the feature extraction and downstream tasks, allowing for greater flexibility in training at a lower compute cost.

---

[*]Both Monica I. and Neha J. contributed equally as first authors

Table 1: Examples of significant MHC gene-to-disease associations for select autoimmune diseases

| Disease | AUC | Sig. Genes | Top 2 Genes | P-Values |
|---------|-----|------------|-------------|----------|
| T1D | 0.55 (0.01) | 60 | *HLA-DQB1-AS1* | 3.32 x $10^{-48}$ |
| | | | *HSPA1B* | 6.39 x $10^{-27}$ |
| PSO | 0.59 (0.01) | 78 | *PSORS1C1* | 1.07 x $10^{-140}$ |
| | | | *TNF* | 4.24 x $10^{-128}$ |

## 2 METHODS

We can define the variant-to-gene bipartite knowledge graph $\mathcal{G}$, with the adjacency matrix $\boldsymbol{A} \in \{0,1\}^{m \times n}$, where $m$ is the number of variants and $n$ is the number of genes. Each row of $\boldsymbol{A}$ is duplicated to reflect the structure of the input space (two features per variant), resulting in updated adjacency matrix $\boldsymbol{A}' \in \{0,1\}^{2m \times n}$. We define $\boldsymbol{x} \in \{0,1\}^{2m}$ as the input and introduce a noise vector $\boldsymbol{z} \sim \mathcal{N}(0,1) \in \mathbb{R}^{2m}$ to distort the input by a factor of $\gamma$ such that $\boldsymbol{x}' = \boldsymbol{x} + \gamma \boldsymbol{z}$. We define GeneDAE below, such that $\boldsymbol{g} \in \mathbb{R}^n$ is the hidden layer with $n$ gene embedding values, $\boldsymbol{W}_1$ and $\boldsymbol{W}_2$ are the encoder and decoder weight matrices, $\boldsymbol{b}$ and $\boldsymbol{c}$ are bias vectors, and $\hat{\boldsymbol{x}}$ is the output. The objective was to minimize the binary cross entropy loss $L(\hat{\boldsymbol{x}}, \boldsymbol{x})$ over all training examples.

$$\boldsymbol{g} = ReLU((\boldsymbol{W}_1 \odot \boldsymbol{A}')\boldsymbol{x}' + \boldsymbol{b}) \quad \text{and} \quad \hat{\boldsymbol{x}} = \sigma((\boldsymbol{W}_2 \odot \boldsymbol{A}'^{\top})\boldsymbol{g} + \boldsymbol{c}) \tag{1}$$

We created an adjacency matrix using variant-to-gene annotations for the major histone compatibility (MHC) genome region using ANNOVAR Yang & Wang (2015), where $m = 8,817$, $n = 239$.

## 3 RESULTS

**Gene-Disease Associations** After training, to explore the relevance of each gene embedding for select autoimmune diseases, we extracted $\boldsymbol{G} \in \mathbb{R}^{s \times n}$ for $s$ samples and ran t-tests comparing mean gene values between cases and controls for each disease with Bonferroni correction ($\alpha = 0.01$). Examples of our findings for Type 1 Diabetes (T1D) and Psoriasis (PSO) are described in Table (1). Not only did we validate existing research on established associations between genes and diseases, as in the case of *PSORS1C1* and *TNF* for PSO or *HLA-DQB1-AS1* for T1D, we also provide some evidence that our approach may enable discovery of new gene-disease associations. For example, *HSPA1B* is a gene encoding a heat shock protein that was previously associated with type 2 diabetes (Buraczynska et al., 2009), but there is no literature supporting the role of this gene in T1D. While other studies have suggested that heat shock proteins play a role in T1D etiology (Moin et al., 2021), our study is the first-ever to suggest a significant association between this gene and T1D. We also visualized the overlap in significant genes between the diseases (Appendix A.2). For example, *HLA-DQA2* is significant for both multiple sclerosis (MS) and PSO, and *LY6G5B*. This suggests that GeneDAE gene embeddings might be used to explore similarities between diseases in terms of their gene-to-disease associations.

**Disease Prediction Performance** We then used the gene embeddings to run multivariate binary classification tasks to predict autoimmune disease risk. The test AUCs ranged from 0.55-0.61 (Table 2 in Appendix A.2). This is a significant result given that only 0.25% of the genome was used, suggesting potential for better performance if we to adapt GeneDAE to genome-wide data beyond the MHC region. The test AUCs and standard error values for T1D and PSO are shown in Table 1.

## 4 CONCLUSION

In conclusion, we demonstrated that GeneDAE extracts interpretable gene embeddings from population-level genotype variant data, offering a potentially useful tool for the discovery or analysis of gene-to-disease associations in genomics research. In future work, we can expand the gene embedding dimensionality to 2 or 3 nodes per gene to enable the embeddings to capture more information in each embedding. We also plan to further validate our approach through more extensive analyses of gene-to-disease association results for additional diseases.

ACKNOWLEDGEMENTS

We acknowledge the members of our Georgia Tech research team for providing constructive feedback in support of this work. In particular, we thank Aleevelu Raparti and Joseph Tsenum for their efforts in support of this work.

URM STATEMENT

The authors acknowledge that Monica I., Neha J., Karan S., Andrew H., and May D. W. meet the URM criteria of ICLR 2023 Tiny Papers Track.

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

## A APPENDIX

### A.1 METHODS - ADDITIONAL DETAILS

#### A.1.1 GENEDAE MODEL ARCHITECTURE AND IMPLEMENTATION DETAILS

The objective for training GeneDAE was to minimize the binary cross entropy loss $L(\hat{x}, x)$ over all training examples $s$.

$$Loss = -\frac{1}{s}\frac{1}{2m}\sum_{i=1}^{s}\sum_{j=1}^{2m}(x_{i,j}\log(\hat{x}_{i,j}) + (1 - x_{i,j})\log(1 - \hat{x}_{i,j})) \qquad (2)$$

GeneDAE was trained using a 70/20/10 train/val/test split with Adam optimizer using a learning rate of 0.0001. We used $\gamma = 0.3$ as the noise factor. Neither dropout nor regularization were used due to the already-sparse architecture. The model converged within 60 epochs. Because GeneDAE is modeled based on a bipartite variant-to-gene knowledge graph with many-to-many connections, it can be used independently for individual genes or gene sets. In our example, we used 8,817 variants and 239 genes in the MHC region, but the GeneDAE framework enables the flexibility to

select just one of those genes in the graph and model the neural network from just the variant-to-gene connections for that gene (see Appendix A.2). This enables the flexibility to train the model sequentially on individual genes even in the absence of extensive computational resources. Likewise, it also allows for the flexibility to expand the feature space and learn embeddings for larger gene sets at once.

### A.1.2 Genotype Data and Pre-Processing, and Variant-to-Gene Annotations

We used the UK Biobank dataset (Bycroft et al., 2018), which comprises population-level genotype and phenotype data on over 502,000 middle aged individuals. The genotype dataset was obtained using the UK Biobank Axiom array chip, and 8,817 variants were available in the MHC region on chromosome 6, spanning nearly 5 million base pairs ranging from position 28,477,797 to position 33,448,355. Each genotype variant originally had an ordinal encoding as 0, 1, and 2 for each individual based on the number of copies of the reference allele. This was then mapped into a binary encoding comprising two features per variant, where [0,0], [1,1], and [1,0] represent having zero, two, or one copies respectively. This resulted in a total of 17,634 input features. We filtered individuals by including just one ancestry group, as is commonly done in genomic studies, for a total of around 470,000 individuals.

To obtain adjacency matrices, we used the genomic annotation software ANNOVAR (Yang & Wang, 2015) to identify genes located close to each of the 8,817 MHC variants of interest. For each variant, we identified any genes for which the variant was exonic (in a coding region of the gene), intronic (in a non-coding region of the gene), splicing (intronic but within 2 base pairs of a splicing junction in the gene), in the 5' or 3' untranslated regions (UTRs), within 1k base pairs upstream or downstream of the transcription start or stop sites, respectively, or located in an intergenic region but relatively close to the gene. We also included variants in transcribed but non-coding RNAs (ncRNAs). A 17,634 x 239 binary adjacency matrix was then created using these connections, whereby each row in the matrix was duplicated to reflect the binary encoding of the 8,817 input genotypes. Based on these annotations, any given variant can be mapped to one or more genes, and any given gene can be mapped to one or variants, resulting in a complex bipartite graph structure in the adjacency matrix. The number and distribution of MHC variants associated with each gene based on the annotations is provided in Figure 2 in Appendix A.2.

### A.1.3 Binary Classification Tasks Implementation Details

For classification tasks, we used Python scikit-learn logistic regression with the 'lbfgs' solver and no regularization. Weighted binary cross-entropy loss was used to correct for class imbalance. Case-control labels for each disease were obtained from diagnostic ICD-10 codes. We used a 70/30 train/test split and ran 10 trials for each analysis. The features were min-max scaled prior to training.

### A.2 Results - Additional Details

The GeneDAE binary cross entropy log loss trajectory for train and validation is in Figure 1. The histogram depicting the connectivity of the variant-to-gene adjacency matrix is shown in Figure 2. A visualization of the variant-to-gene connectivity and flexibility of the GeneDAE architecture is shown in Figure 3. The diseases evaluated for binary classification tasks in Table 2 included Type 1 Diabetes (T1D), Psoriasis (PSO), Multiple Sclerosis (MS), and Ankylosing Spondylitis (AS). The most significant gene-to-disease association results from the t-test analysis between gene embeddings and case-control status for three autoimmune diseases (T1D, PSO, and MS) are displayed in the heatmap in Figure 4, where it is clear that some highly significant genes overlap between different autoimmune diseases. For example *HLA-DQA2* is significantly associated with both MS and PSO, and *LY6G5B* is significant for all three of the autoimmune diseases.

Table 2: Summary of binary classification task results for autoimmune disease risk prediction

| DISEASE | CASES | CONTROLS | PREVALENCE (%) | TEST AUC (95%CI) |
|---------|-------|----------|----------------|------------------|
| T1D | 6,936 | 424,174 | 1.64% | 0.55 (0.01) |
| PSO | 11,589 | 419,521 | 2.76% | 0.59 (0.01) |
| MS | 2,070 | 429,040 | 1.64% | 0.61 (0.02) |
| AS | 3,504 | 427,606 | 0.82% | 0.59 (0.01) |

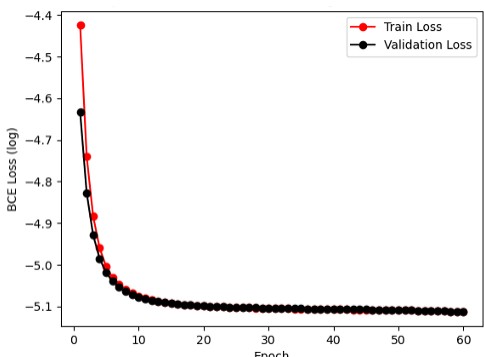

Figure 1: Epochs vs. log binary cross entropy (BCE) loss for GeneDAE training for 60 epochs.

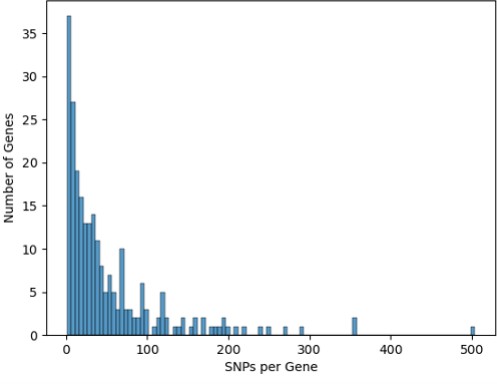

Figure 2: Histogram of SNPs per gene in GeneDAE adjacency matrix.

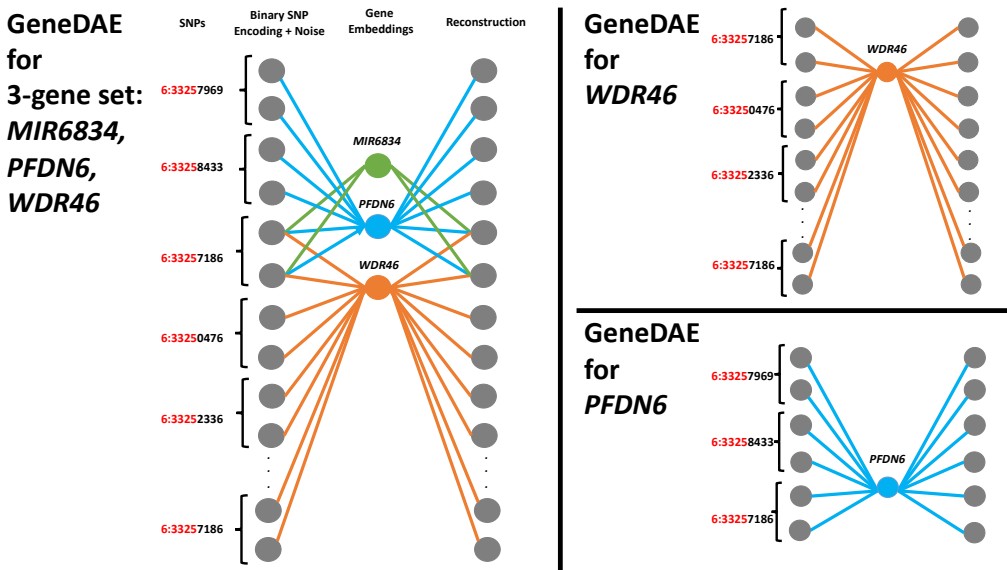

Figure 3: Example visualization of GeneDAE model flexibility enabling independent setup and training to derive gene embeddings for individual genes or gene sets.

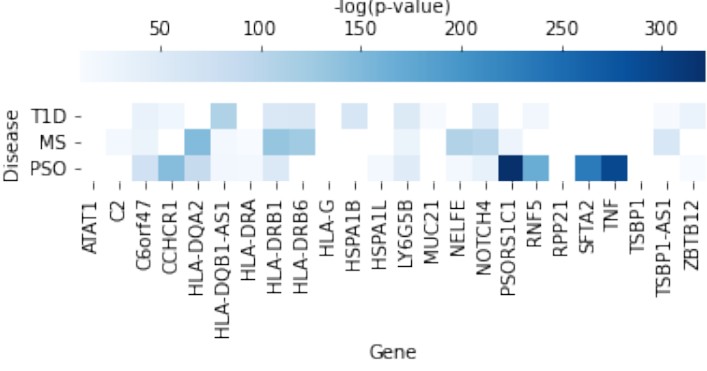

Figure 4: Gene to disease association p-values from t-tests.

