# OpenReview forum: "GeneDAE: A Sparse Denoising Autoencoder for Deriving Interpretable Gene Embeddings"
_ICLR.cc/2023/TinyPapers — Submitted to Tiny Papers @ ICLR 2023_

### Official Review · Reviewer_jjHB · 2023-03-29

**Confidence:** 4

**Summary Of Contributions:**

The paper introduces GeneDAE a sparse denoising autoencoder which uses an adjacency matrix to constrain the connectivity of the encodings and decodings in a model on variant-to-gene annotations connecting variants to nearby genes. This concept is used to extract multi-purpose embeddings of genes that can be used in downstream applications.

**Rating:**

Great Start (GS): a submission which meets some of the reviewing criteria but has room for improvement

**Strengths And Weaknesses:**

**Strengths**

- The paper is well written.  The method is concise and easy to follow.
- The paper shows the ability to extract interpretable gene embeddings from population-level genotype variant data, offering a potentially useful tool for the discovery or analysis of gene-to-disease associations in genomics research.
- The presented results suggest a significant association between the HSPA1B gene and T1D

**Weaknesses**

- While it seems to work well, the paper does not explain the rationale behind some of the choices made. Can this method be used in other model types such as Graph Models, considering the use of adjacency matrixes?
- There is no comparison with other methods, if any. This will be important to further validate the paper's claims.
- There is no information about the reproducibility of the results.

**Suggested Changes:**

- The authors should include information about reproducibility in the paper.

---

### Official Review · Reviewer_g4fE · 2023-03-30

**Confidence:** 4

**Summary Of Contributions:**

The authors propose a sparse denoising autoencoder that extracts gene representations from large-scale population-level genotype data, which can be used to identify gene-to-disease associations. The autoencoder connections are modeled on a bipartite biological knowledge graph that connects individual variants to nearby genes.

**Rating:**

Clear, Correct, and Reproducible (CCR): a submission which meets the reviewing criteria

**Strengths And Weaknesses:**

The graph input enables each node in the hidden layer to be used as an interpretable, multi-purpose gene embedding based on information from close variants most likely to impact gene function. The proposed model uses knowledge graphs (already proposed) in an autoencoder to extract multi-purpose embeddings of genes that can then be used in downstream applications. It also enables gene embeddings to be derived independently for each gene or gene set of interest and decouples the feature extraction and downstream tasks, achieving lower computational costs. The proposed model and experimental setup are clear and reproducible and the achieved results seem significant/interesting.


**Suggested Changes:**

Some main results from experiments (gene-disease associations/disease prediction) should be in the main text (currently all results are in the Appendix).
Would be interesting to include comparisons with other benchmark models.

---

### Meta-Review · Area_Chair_rgX6 · 2023-04-04

**Recommendation:** Invite to present
**Confidence:** 4

**Metareview:**

Good paper with all reviewers arguing for acceptance. It could be better if suggestions from reviewers could be incorporated into the manuscript.

**Summary:**

The authors suggest a sparse denoising autoencoder for extracting gene representations from extensive population-level genotype data, enabling the detection of associations between genes and diseases. The paper is well-written and the finding is clear and interesting.

**Reason For Not Giving A Higher Recommendation:**

Lack of comprehensive discussion on the methodological design. Lack of empirical comparison with other methods.

**Reason For Not Giving A Lower Recommendation:**

N/A

---

### Decision · Program_Chairs · 2023-04-08

Invite to present